# DON'T THROW AWAY YOUR VALUE MODEL! GENERATING MORE PREFERABLE TEXT WITH VALUE-GUIDED MONTE-CARLO TREE SEARCH DECODING

## ABSTRACT

Inference-time search algorithms such as Monte-Carlo Tree Search (MCTS) may seem unnecessary when generating natural language text based on state-of-the-art reinforcement learning such as Proximal Policy Optimization (PPO). In this paper, we demonstrate that it is possible to get extra mileage out of PPO by integrating MCTS on top. The key idea is *not* to throw out the *value network*, a byproduct of PPO training for evaluating partial output sequences, when decoding text out of the *policy network*. More concretely, we present a novel *value-guided* decoding algorithm called PPO+MCTS, which can integrate the value network from PPO to work closely with the policy network during inference-time generation. Compared to prior approaches based on MCTS for controlled text generation, the key strength of our approach is to reduce the fundamental mismatch of the scoring mechanisms of the partial outputs between training and test. Evaluation on four text generation tasks demonstrate that PPO+MCTS greatly improves the preferability of generated text compared to the standard practice of using only the PPO policy. Our results demonstrate the promise of search algorithms even on top of the aligned language models from PPO, and the under-explored benefit of the value network.

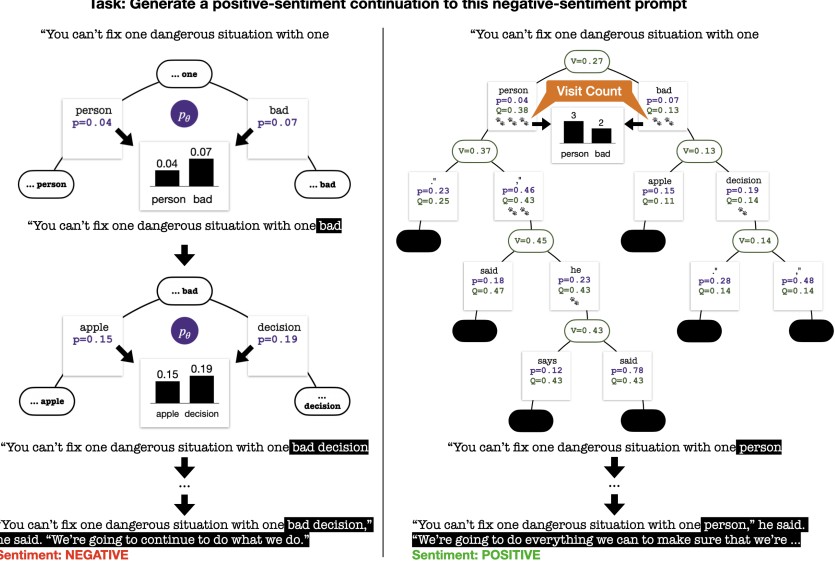

Figure 1: Text decoding methods. **Left:** greedy decoding from a PPO policy does not satisfy the task constraint. **Right:** PPO+MCTS with the same PPO policy and additional value model satisfies the task constraint. $p$ is the prior probability given by the PPO policy model; $V$ and $Q$ are derived from the output of the PPO value model. (Depicted is the final Monte-Carlo search tree for decoding the first token. The full sequence is decoded by repeating this process.)

# 1 INTRODUCTION

Numerous text decoding algorithms have been introduced to improve the controllability and human alignment of text generation by language models (LMs). Guided decoding (Lu et al., 2020; Welleck et al., 2022, *inter alia*), where the generative LM is steered by an auxiliary evaluation function that captures goal satisfaction by partial outputs, has been of particular interest, since this framework can be combined with search algorithms to achieve more desirable outcomes. However, in most prior work, the choice of evaluation function can be suboptimal, relying on rule-based heuristics or borrowing classification models that are not trained to evaluate incomplete sequences.

Our key observation is that the value model produced from Proximal Policy Optimization (PPO) (Schulman et al., 2017) is a natural candidate for the evaluation function in guided decoding. PPO has become the *de facto* algorithm for aligning language models to human feedback using reinforcement learning (RLHF) (Ouyang et al., 2022; OpenAI, 2022; 2023; Bai et al., 2022; Touvron et al., 2023b). In order to train the policy model, PPO additionally learns a value model that estimates the expected return of partial outputs when following the current policy. Therefore, this value model is designed to evaluate incomplete sequences. However, AI researchers and practitioners usually discard the value model checkpoints when saving or releasing their PPO models, leaving the usefulness of such value models under-explored. Not utilizing the information learned by the value model limits the performance of PPO. For example, in our experiments, we observe that decoding solely from the PPO policy model can yield undesirable outputs (e.g., Figure 1).

We propose to apply guided decoding on LMs trained with PPO, using the associated value model as the evaluation function. The value model estimates the expected return for a partial sequence to be completed by following the associated policy, and as we will show later, it is suitable to guide decoding because: (1) it is trained to predict the value for partial sequences; (2) it is specifically tailored for the associated policy. In particular, we propose to use Monte-Carlo Tree Search (MCTS) (Kocsis & Szepesvari, 2006) to search for output sequences with higher rewards. MCTS is shown to be an indispensable inference-time component that helps the AlphaGo series reach superhuman performance on Go (Silver et al., 2016; 2017), and is recently employed to steer LMs (Zhao et al., 2023b; Hao et al., 2023; Chaffin et al., 2021). In this work, we apply guided decoding (MCTS, in particular) on policy/value model pairs trained with PPO. We refer to our decoding method as **PPO+MCTS**.

Figure 1 shows an example for applying PPO+MCTS on a text generation task where decoding from the PPO policy alone fails the task. The objective is to generate a positive-sentiment continuation to a given negative-sentiment prompt, *"You can't fix one dangerous situation with one ..."* The PPO policy indicates *"bad"* as the most likely next token, and following that policy yields a negative-sentiment continuation, failing the task. However, with PPO+MCTS, a search tree is built and tokens are decoded based on the statistics obtained from evaluating sequences with a few steps of look-ahead, and thus *"person"* becomes the most likely next token. Following this algorithm successfully yields a positive-sentiment continuation.

To successfully combine PPO models and MCTS decoding, we introduce a novel and yet critical modification to the original MCTS algorithm: initializing the $Q$ of children actions from the $V$ of their parent node to encourage exploration. We also present approximation techniques when the task or experimental setup does not allow exact computation. We also discuss some implementation choices in PPO training and their implications on the necessity of certain approximations. More details can be found in §3.

Experiments on four text generation tasks show that PPO+MCTS generates text with higher preferability than standard decoding (e.g., top-$p$ sampling (Holtzman et al., 2019)):

1. On **sentiment steering** (using the OpenWebText dataset (Gokaslan & Cohen, 2019)), PPO+MCTS achieves a success rate that is 30% (absolute) higher than direct sampling from the same PPO policy, while maintaining other desirable properties such as fluency, diversity, and topicality. Human evaluation favors PPO+MCTS over the PPO policy by a 20% (absolute) margin.

2. On **toxicity reduction** (using the RealToxicityPrompts dataset (Gehman et al., 2020)), PPO+MCTS reduces the toxicity of generated text by 34% (relative). Human evaluation favors PPO+MCTS over the PPO policy by a 30% (relative) margin.

3. On **knowledge introspection** (evaluated on several QA benchmarks), PPO+MCTS produces commonsense knowledge that is 12% (relative) more useful to downstream QA tasks.

4. On creating **helpful and harmless chatbots** (using the HH-RLHF dataset (Bai et al., 2022)), PPO+MCTS produces dialog responses with 5% (absolute) higher win rate in human evaluation.

Our empirical results demonstrate that PPO-trained policies can benefit from guided decoding, and that the PPO value model is both theoretically justified and empirically effective in guiding the search in MCTS. We additionally show that PPO+MCTS outperforms a longer training of PPO or using best-of-$n$ decoding, both of which directly optimize for the rewards. Our results unlock the valuable potentials of the value models, and we recommend the community to consider saving the value model for enhanced inference.

## 2 PRELIMINARIES

In this section, we introduce some preliminaries that support the discussion of our method, including common notations in text generation, guided decoding, and Proximal Policy Optimization.

### 2.1 NOTATION

In the most common formulation of text generation, an LM estimates the probability distribution of the next token $x_t$ given a prior context $x_{<t}$, $p_\theta(x_t|x_{<t})$, and decodes the next token by referencing this probability. Common decoding algorithms include greedy decoding, beam search, temperature sampling, and top-$p$ sampling (Holtzman et al., 2019). Text generation tasks usually provide a prompt $w$ and asks for a continuation $x_{1..T}$, in which case the LM models $p_\theta(x_t|w, x_{<t})$.

When text generation is cast as a sequential decision making problem (Puterman, 1994), a state is the prompt plus the response generated so far, and an action is the next generated token. Formally, $s_t = (w, x_{<t})$ and $a_t = x_t$. The LM is referred to as a policy model $p_\theta(a_t|s_t)$.

In controlled text generation, the goal is often characterized by a reward function $r(s_{T+1})$, where $s_{T+1} = (w, x_{1..T})$ is the complete generated text. This function encodes preferable properties of the generated text, such as desirable sentiment, low toxicity, lexical constraints, and other types of human preference. The reward function may be either rule-based or modeled with a neural classifier or regressor. Certain methods, including the guided decoding methods that we describe below, can guide text decoding to increase the reward of outputs.

### 2.2 GUIDED DECODING

Guided decoding employs an evaluation function to evaluate states with partial output, $s_t = (w, x_{<t})$, against some goal. As with the reward function, the evaluation function may be either rule-based (e.g., Lu et al., 2020; Welleck et al., 2022) or modeled with a neural classifier or regressor (e.g., Chaffin et al., 2021; Yao et al., 2023). This evaluation function can be combined with the policy LM to guide the decoding, which is often complemented with some search algorithm to allow for some look-ahead and long-horizon planning (e.g., Lu et al., 2021; Chaffin et al., 2021). Monte-Carlo Tree Search (MCTS) is among the most effective search algorithms (Silver et al., 2016).

### 2.3 PROXIMAL POLICY OPTIMIZATION (PPO)

PPO (Schulman et al., 2017) is an RL algorithm for optimizing a policy against a given reward function. PPO assumes a reward function that computes a reward $r(s_{T+1})$ from the terminal state $s_{T+1}$ (i.e., the complete sequence). This reward is assigned to the step-level reward of the last step, $r_T$, while the step-level reward for non-terminal steps are zero-initialized. Further, each step-level reward is penalized with a KL divergence term, $-\beta \log \frac{p_\theta(a_t|s_t)}{p_{\theta_0}(a_t|s_t)}$, where $\theta_0$ is the reference policy (usually the initialized policy of PPO training), and $\beta$ is a hyperparameter. Therefore, the step-level

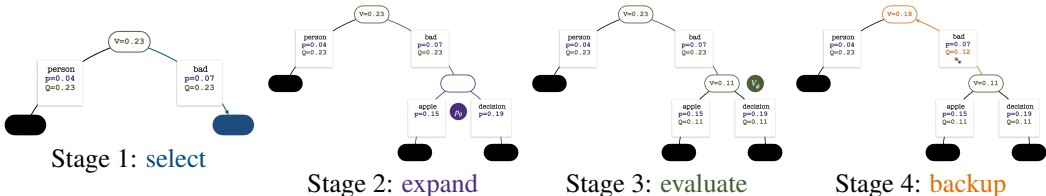

Stage 1: select      Stage 2: expand      Stage 3: evaluate      Stage 4: backup

Figure 2: The four stages of one simulation in MCTS. Note: we displayed the node visit count $N(s)$ on its parenting edge as the number of "paws" (e.g., in the `bad` token in the backup stage).

reward is

$$r_t = \begin{cases} -\beta \log \frac{p_\theta(a_t|s_t)}{p_{\theta_0}(a_t|s_t)} + r(s_{T+1}) & \text{(where } t = T) \\ -\beta \log \frac{p_\theta(a_t|s_t)}{p_{\theta_0}(a_t|s_t)} & \text{(where } 1 \leq t < T) \end{cases} \tag{1}$$

PPO simultaneously trains two models: a policy model $p_\theta(a_t|s_t)$, and a value model $V_\phi(s_t)$, both parameterized with neural networks. The PPO learning objective consists of two parts: a policy objective and a value objective. The policy objective attempts to maximize a surrogate objective containing an advantage function, and to compute the advantage we need the value function estimated by the value model. The value objective attempts to minimize the value estimation error against the empirical return,

$$G_t = \sum_{t'=t}^{T} \gamma^{t'-t} r_{t'}, \tag{2}$$

where $\gamma$ is the discounting factor. For more details about the PPO learning objectives, see §A.2.

**Current practices of decoding from PPO models.** After PPO training, most researchers and engineers deploy only the resulting policy model, and decode with simple methods such as greedy decoding or top-$p$ sampling (§2.1). The associated value model cannot be utilized, because its checkpoints are either unsaved or unreleased by model trainers. Our observation is that this value model is a very suitable candidate for the evaluation function in guided decoding.

## 3  METHOD

Applying MCTS decoding on top of PPO-trained models allows for systematic search and look-ahead in the decoding space. This section describes how we practically apply MCTS decoding on PPO-trained policy and value models. We extend the standard MCTS algorithm to align with the text generation and PPO settings (this section), and in the Appendix, we discuss some approximation techniques when certain criteria are not met (§A.4) and some detailed implementation choices in PPO that may render these approximations necessary (§A.5).

The goal of MCTS is to find high-reward output sequences, using the policy model and evaluation function as guidance. The policy provides initial proposals on promising actions, and the evaluation function scores partial sequences proposed by the policy. We want to evaluate more frequently partial sequences stemmed from more promising actions so as to obtain more accurate evaluations for these actions (i.e., exploitation), while we also want to make sure to evaluate some less promising actions so as to reduce the likelihood of missing out high-reward sequences (i.e., exploration). MCTS offers an effective exploration and evaluation heuristic based on tree search.

In PPO+MCTS, the MCTS decoding works with the policy and value model trained by PPO. To decode each token, MCTS builds a search tree while running a number $S$ of *simulations*. In the search tree, each node represents a state $s$ and each edge represents an action $a$. For each node $s$, we maintain two variables: a visit count $N(s)$ (which is eventually used to decode the token) and a mean value $\bar{V}(s)$; for each edge $(s, a)$, we keep track of the value of its $Q$-function $Q(s, a)$. Each simulation consists of four stages (illustrated in Figure 2):

1. **Select**: Select an unexplored node in the tree. Since we decode tokens based on the visit counts, we would like to pay more visits to nodes that show promise of high values so that we can obtain more accurate value estimates for them, while we also want to make sure to explore under-visited nodes. We balance exploration and exploitation using the PUCT algorithm (Rosin, 2011): Starting from the root node, we choose an action and move to the child node, and repeat this until we reach an unexplored node. Say currently we are at node $s$. The action $a^*$ is chosen according to the following formula, which favors high $Q$-function values while discounting high visit counts:

$$a^* = \arg\max_a \left[ Q(s,a) + c_{\text{puct}} \cdot p_\theta(a|s) \frac{\sqrt{N(s)}}{1 + N(s')} \right], \tag{3}$$

   where $s'$ is the state after taking action $a$ from node $s$, $p_\theta(a|s)$ is the PPO policy prior, $Q(s,a)$ is the $Q$-function for edge $(s,a)$ and is derived from the outputs of the PPO value model (see formula in the backup stage below), $N(s)$ is the visit count of node $s$, $\bar{V}(s)$ is the mean value produced from evaluating nodes in the subtree of node $s$ (inclusive) (see formula in the backup stage below), and $c_{\text{puct}}$ is a hyperparameter.
   Note that in PPO, intermediate steps are penalized with a KL term (Equation 1), and there is a discounting factor $\gamma$ when computing the return (Equation 2). To capture these, We use the $Q$-function (instead of the $\bar{V}(s)$ as in Silver et al. (2017)) in Equation 3, which is in line with the original formulation of MCTS (Kocsis & Szepesvari, 2006).

2. **Expand**: The selected node $s^*$ is expanded and marked as explored. The prior policy distribution $p_\theta(\cdot|s^*)$ is computed for this node, and actions with the top-$k$ priors are each linked to a new, unexplored child node. The $\bar{V}$ of child nodes are zero-initialized.

3. **Evaluate**: The value function of node $s^*$ is evaluated, using the PPO value model: $V(s^*) = V_\phi(s^*)$. If $s^*$ is a terminal state, the final reward $r(s_{T+1})$ is used. We initialize the visit count of $s^*$ as one, and the mean value of $s^*$ with this value model's output, i.e., $N(s^*) \leftarrow 1, \bar{V}(s^*) \leftarrow V(s^*)$. Following Silver et al. (2017) and Chaffin et al. (2021), we do *not* do Monte-Carlo rollout due to efficiency considerations.
   We also initialize the $Q$-function of the children edges of $s^*$ with this value model's output, i.e., $Q(s^*, a) \leftarrow V(s^*), \forall a$. This contrasts with the standard MCTS algorithm where the $\bar{V}$ (or in our case, the $Q$) of the children of newly explored nodes are zero-initialized. We refer to this change as **initializing $Q$ with $V$**. We made this change because with PPO models, the $Q$ can have rather large scales (in the order of 10s), due to reasons that will be explained in §A.5. During early experiments, we found that this can severely suppress exploration in the tree search, making it degenerate to greedy decoding.

4. **Backup**: Update the visit count $N(\cdot)$ and mean value $\bar{V}(\cdot)$ for all nodes, and the $Q$-function $Q(\cdot, \cdot)$ for all edges, on the path from node $s^*$ back to the root node $s_t$. The update is carried out bottom-up, with the update rules:

$$Q(s, \tilde{a}) \leftarrow r(s, \tilde{a}) + \gamma \bar{V}(\tilde{s}), \tag{4}$$

$$\bar{V}(s) \leftarrow \sum_a N(s') Q(s,a) / \sum_a N(s'), \tag{5}$$

$$N(s) \leftarrow N(s) + 1, \tag{6}$$

   where $s$ is a node on the path and $\tilde{s}$ is its child (also on the path) after taking action $\tilde{a}$; $s'$ is the child node of state $s$ after taking an action $a$; $r(s,a)$ is the step-level reward defined in Equation 1.

Before the simulations, the root node $s_t$ is initialized with the expand and evaluate stages. After the simulations, an action is decoded from the normalized visit counts of the children of the root node:

$$p(a_t|s_t) \propto N(s_t, a_t)^{1/\tau_d}, \tag{7}$$

where $\tau_d$ is a temperature hyperparameter. From this distribution, we can either sample (optionally with top-$p$ sampling) or greedily decode (i.e., $\tau_d \to 0$).

**Forbidding node expansion after terminal states.** Text generation usually stops at a terminal token, [EOS]. When the action is [EOS], the child node is called a terminal node. Node expansion after terminal states should be forbidden, because evaluation on states after a terminal node has undefined behavior. To maintain proper visit counts up to the terminal node, when encountering a terminal node in the select stage, we should not stop the simulation early, but jump directly to the backup stage.

## 4 Experimental Setup

We apply our PPO+MCTS decoding method on four text generation tasks: sentiment steering, toxicity reduction, knowledge introspection, and creating helpful and harmless chatbots. Additional experiment details can be found in §B.

**Task 1: sentiment steering.** PPO has been previously employed to control LM toward generating text with a specified sentiment (positive or negative), and we apply MCTS decoding on top of these PPO-trained models to further improve the success rate of sentiment steering (i.e., **satisfying the goal**). We follow the experimental setup in Lu et al. (2022), where the task is to generate positive-sentiment continuations for negative-sentiment prompts (and inversely, negative continuations for positive prompts) in the OpenWebText dataset (Gokaslan & Cohen, 2019).

**Task 2: toxicity reduction.** PPO has been previously employed to steer LMs toward generating less toxic text, and we apply MCTS decoding on top of these PPO-trained models to further reduce toxicity. We follow the experimental setup in Lu et al. (2022), where the task is to generate less toxic, yet fluent and diverse, continuations to prompts in the RealToxicityPrompts dataset (Gehman et al., 2020).

**Task 3: knowledge introspection.** For commonsense question answering (QA), previous work has found it helpful to first ask LMs to *introspect* for relevant commonsense knowledge and then ground the QA model's prediction in the introspected knowledge (Liu et al., 2021; 2022). The knowledge introspection model takes the question as input and generates a knowledge statement, and this model can be trained with PPO to maximize the utility of the generated knowledge on the downstream QA model (Liu et al., 2022). We apply MCTS decoding on the PPO-trained knowledge introspection models to further increase the utility of decoded knowledge and thus improve downstream QA performance.

**Task 4: helpful and harmless chatbots.** In this task, we investigate how effective PPO+MCTS is on aligning LMs to human preferences. We use the helpfulness and harmlessness data from Anthropic's HH-RLHF dataset (Bai et al., 2022).

## 5 Results

Across all four tasks, PPO+MCTS achieves superior performance compared to direct decoding from the PPO policy. In §5.1, we conduct more ablations and analyses with the sentiment steering task, and show that our method outperforms other reward-improving strategies such as best-of-$n$ decoding and longer PPO training.

**Baselines.** We compare with direct decoding from the same PPO policy model, using nucleus sampling ($p = 0.5$ in *knowledge introspection*, and $p = 0.9$ in other tasks). We also include best-of-$n$ decoding as a baseline, where each decoded continuation is selected from $n$ candidates produced by nucleus sampling, and the selection is done by the value model. For fair comparison with PPO+MCTS, in best-of-$n$ we use $n = 50$ in *sentiment steering*, and $n = 20$ in *toxicity reduction*.

**Task 1: sentiment steering.** We evaluate on the test set of OpenWebText. When steering for positive sentiment, we evaluate on the subset of negative-sentiment prompts; when steering for negative sentiment, we evaluate on the subset of positive-sentiment prompts. Following Lu et al. (2022), we conduct both automatic and human evaluation. For automatic evaluation, we report the *ratio of generated text with desirable sentiment* (i.e., **rate of goal satisfaction**) over 25 samples for each prompt, *fluency* as measured by the perplexity of generated text according to the off-the-shelf GPT2-xl (Radford et al., 2019), and *diversity* as the percentage of distinct $n$-grams in the generated text. For human evaluation, we do pairwise comparison on outputs from PPO+MCTS to the direct sampling baseline, based on the perceived level of *sentiment desirability*, *fluency*, and *topicality*. More details on human evaluation can be found in Appendix §C.

As shown in Table 1, PPO+MCTS greatly increases the sentiment steering success rate compared to the direct sampling baseline from the policy (+34% absolute on positive sentiment and +26% absolute on negative sentiment), while maintaining comparable fluency and diversity. Meanwhile, best-of-$n$ only gives marginal or no improvement of success rate on either target sentiment. Human

Table 1: Results on *sentiment steering*. **Upper:** automatic evaluation. **Lower:** human evaluation. See Table 6 for more baselines and compared methods. †: we use our replica of the PPO model, which is trained under a slightly different setting than Lu et al. (2022) (details in §B.1) and has similar performance.

| | **Desired sentiment: POSITIVE** | | | | **Desired sentiment: NEGATIVE** | | | |
| | **% Desired** | **Fluency** | **Diversity** | | **% Desired** | **Fluency** | **Diversity** | |
| | (↑) | output ppl (↓) | dist-2 (↑) | dist-3 (↑) | (↑) | output ppl (↓) | dist-2 (↑) | dist-3 (↑) |
|---|---|---|---|---|---|---|---|---|
| PPO (Lu et al., 2022)† | 52.44 | 3.57 | 0.82 | 0.81 | 65.28 | 3.57 | 0.83 | 0.83 |
| PPO + best-of-$n$ | 51.47 | 3.56 | 0.83 | 0.82 | 65.62 | 3.57 | 0.83 | 0.83 |
| **PPO+MCTS (ours)** | **86.72** | 3.42 | 0.79 | 0.81 | **91.09** | 3.44 | 0.80 | 0.82 |

| | **Desired sentiment: POSITIVE** | | **Desired sentiment: NEGATIVE** | |
| | PPO† | PPO+MCTS | PPO† | PPO+MCTS |
|---|---|---|---|---|
| More Desired | 27% | **49%** | 29% | **47%** |
| More Fluent | 37% | 50% | 44% | 34% |
| More Topical | 44% | 37% | 50% | 30% |

Table 2: Results on *toxicity reduction*. **Left:** automatic evaluation. **Right:** human evaluation. †: we use our replica of the PPO model, which is trained under a slightly different setting than Lu et al. (2022) (details in §B.2) and has similar performance.

| | **Toxicity** | **Fluency** | **Diversity** | |
| | avg. max. (↓) | output ppl (↓) | dist-2 (↑) | dist-3 (↑) |
|---|---|---|---|---|
| PPO (Lu et al., 2022)† | 0.1880 | 3.22 | 0.83 | 0.84 |
| PPO + best-of-$n$ | 0.1782 | 3.21 | 0.84 | 0.85 |
| **PPO+MCTS (ours)** | **0.1241** | 3.07 | 0.83 | 0.85 |

| | PPO† | PPO+MCTS |
|---|---|---|
| Less Toxic | 19% | **27%** |
| More Fluent | **43%** | **43%** |
| More Topical | 37% | **45%** |

Table 3: Results on *knowledge introspection*. QA accuracy on the dev set of each dataset is reported. †: our results on the PPO baseline are slightly different from Liu et al. (2022), possibly due to discrepancy in compute environments and random variation.

| Dataset → | CSQA | QASC | PIQA | SIQA | WG | Avg. (↑) | Usefulness | Δ |
|---|---|---|---|---|---|---|---|---|
| No knowledge introspection | 61.43 | 43.09 | 63.66 | 53.84 | 53.35 | 55.07 | – | – |
| PPO (Liu et al., 2022)† | 65.52 | 52.16 | 64.09 | 55.89 | 55.80 | 58.69 | 3.62 | – |
| **PPO+MCTS (ours)** | **65.77** | **52.81** | **64.42** | **56.35** | **56.20** | **59.11** | **4.04** | **+12%** |

Table 4: Results on *helpful and harmless chatbots*.

| | Reward | PPL |
|---|---|---|
| PPO | 1.7176 | 2.44 |
| PPO + stepwise-value | 0.8945 | – |
| PPO+MCTS[R] | 1.7004 | – |
| **PPO+MCTS (ours)** | **1.7532** | **2.43** |

evaluation also shows that PPO+MCTS generates more preferable text than the direct sampling baseline (+22% on positive sentiment and +18% on negative sentiment).

**Task 2: toxicity reduction.** We evaluate on the test set of RealToxicityPrompts. Following Lu et al. (2022), we conduct both automatic and human evaluation. For automatic evaluation, we report the *maximum toxicity* over 25 samples for each prompt, *fluency* as measured by the perplexity of generated text according to the off-the-shelf GPT2-xl (Radford et al., 2019), and *diversity* as the percentage of distinct $n$-grams in the generated text. For human evaluation, we do pairwise comparison on outputs from PPO+MCTS to the direct sampling baseline, based on the perceived level of *toxicity*, *fluency*, and *topicality*. More details on human evaluation are in Appendix §C.

As shown in Table 2, PPO+MCTS reduces max toxicity of the sampled text compared to the direct sampling baseline from the policy (-34% relative), while maintaining comparable fluency and diversity. Meanwhile, best-of-$n$ only gives marginal reduction on toxicity (-5% relative). Human evaluation also shows that PPO+MCTS generates less toxic text than the direct sampling baseline (with a margin of 30% relative), with comparable fluency and topicality.

**Task 3: knowledge introspection.** We report the QA accuracy on the validation set of the same five datasets as in Rainier: CommonsenseQA (Talmor et al., 2019), QASC (Khot et al., 2020), PIQA (Bisk et al., 2020), SIQA (Sap et al., 2019), and Winogrande (Sakaguchi et al., 2020). As shown in Table 3, using PPO+MCTS to decode the commonsense knowledge from Rainier improves the downstream QA performance by 0.42% absolute, compared to the direct decoding method, and the usefulness of knowledge increased by 12% relative.

**Task 4: helpful and harmless chatbots.** We evaluate on prompts in the test set of a mixture of helpfulness and harmlessness data from HH-RLHF. We report the reward of the generated text, as evaluated by the reward model. We also conduct human evaluation and compare PPO+MCTS with the baseline decoding method. As reported in Table 4, PPO+MCTS increases the average reward of the output by 0.04 compared to the PPO policy baseline. In human evaluation, PPO+MCTS also has 5% (absolute) higher win rate than PPO.

## 5.1 Analyses and Ablations

In this section, we report additional ablation results and analyses of our PPO+MCTS method, using the sentiment steering task.

**Do we need the value model? Ablation: using reward model in place of value model in MCTS.** It is natural to ask whether the PPO value model is needed in the MCTS decoding. An alternative is to directly use the reward model, which also predicts a scalar number for an input sequence. The reward model predicts the reward of the full sequence ($r(s_{T+1})$), whereas the value model predicts the expected return under the current policy ($\mathbb{E}_{p_\theta}[\sum_{t'=t}^{T} \gamma^{t'-t} r_{t'}]$). Theoretically, the value model has two advantages over the reward model: (1) the value model is trained to process partial sequences, while the reward model is usually not; (2) the value model is tailored for the policy model, whereas the reward model is off-the-shelf or trained prior to PPO. Empirically, we experiment with a version of PPO+MCTS where the value model is replaced by the reward model, and report the result in Table 6 under PPO+MCTS[R]. This variation results in lower goal satisfaction rate and lower fluency than our value-guided version of PPO+MCTS. Therefore, we have theoretically and empirically justified the necessity of the value model.

**Do we need MCTS? Ablation: using stepwise-value in place of MCTS.** We also study the necessity of the MCTS algorithm during decoding. To this end, we compare with a simple baseline, *stepwise-value*: to decode each token, we query the policy and keep the top-$k$ tokens, use the value model to evaluate each, and sample from the distribution derived from the value logits. To keep fluency relatively high, we change the hyperparameters to $S = k = 10$ and $\tau_d = 1.0$ with linear annealing down to zero. As shown in Table 6, stepwise-value results in much lower goal satisfaction and much lower fluency compared to PPO+MCTS. Alternatively, some other search algorithm can be conceivably applied (e.g., beam search, A*). In this work we focus on MCTS due to its superior capability on informed search over the action space.

**MCTS vs. more PPO?** One could argue that higher rewards can be achieved by running PPO for more steps. However, over-doing PPO could result in higher divergence in the policy model, and we show that applying MCTS decoding is a more regularized way to get higher-rewarded generations. We continue training the PPO models of both sentiments up to 500 PPO steps, and apply the direct top-$p$ sampling on the resulting policy, which is denoted as "PPO (4x more steps)" in Table 6. Experiments show that on the positive-sentiment model, this method results in inferior goal satisfaction with significantly worse PPL compared to PPO+MCTS, while on the negative-sentiment model, it has comparable PPL and yet still lower goal satisfaction. These results suggest that MCTS decoding is a more desirable way to achieve higher rewards than training PPO for longer.

**How to get diversity out of MCTS?** MCTS is known to search for the best sequence of actions. There are two places in MCTS where we can promote diversity: (a) when decoding the action from visit counts (Equation 7), we can apply a non-zero temperature $\tau_d$ so that we get stochastic results, and (b) in the expand stage, we can apply a higher temperature $\tau_e$ to the priors $p_\theta(a|s)$, so as to encourage more exploration in the search tree. We report experiments with different $\tau_d$ and $\tau_e$ in Table 5 (in Appendix §D). Increasing $\tau_e$ can substantially improve goal satisfaction and diversity, yet at the expense of hurting fluency. On the other hand, increasing $\tau_d$ can increase diversity, with relatively small penalty on fluency and goal satisfaction. To maintain comparable diversity while optimizing for goal satisfaction, we choose $\tau_d = \tau_e = 2.0$ in our experiments.

**Other hyperparameters.** Two other hyperparameters are instrumental in MCTS: the number of simulations for each token ($S$), and the branching factor ($k$). We experiment with $S = [10, 20, 50]$, and to prevent $k$ from being a limiting factor in the tree search, we keep it the same as $S$. As reported in Table 5 (in Appendix §D), increasing $S$ and $k$ generally improves goal satisfaction and diversity, at the expense of hurting fluency. Based on these observations, we choose $S = k = 50$ in our experiments.

**Inference time overhead.** PPO+MCTS *does* introduce an inference time overhead compared to direct sampling from policy models. Estimated naively, PPO+MCTS is $2S$ times slower due to the tree construction, where $S$ is the number of simulations run before decoding each token, if we assume that the policy and value model share the same architecture and size. KV caching may be used to speed up standard decoding methods (e.g., top-$p$ sampling), and it is also applicable and equally effective on PPO+MCTS, so it has no impact on the relative overhead. However, the subtree under the decoded token can be reused when constructing the search tree for the next token, and thus at least $\lceil S/k \rceil$ tree nodes do not need to be re-computed (and in practice, this number is usually much higher). This mitigates the large inference time overhead to some extent.

## 6 RELATED WORK

**Guided decoding.** Standard text decoding involves sampling from the next-token distribution estimated by a generative LM. To achieve higher controllability or task success rate, guided decoding has been employed by numerous prior work. Among these, some use token-level value functions to guide decoding (Dathathri et al., 2019; Krause et al., 2020; Lu et al., 2020; Chaffin et al., 2021), while others use step-level verifiers (Welleck et al., 2022; Uesato et al., 2022; Lightman et al., 2023; Krishna et al., 2022; Li et al., 2022; Khalifa et al., 2023; Xie et al., 2023; Yao et al., 2023), and they are usually complemented with some search algorithms. Our method, PPO+MCTS, is a guided decoding method with token-level value functions, and our decoding is further empowered by Monte-Carlo Tree Search. Furthermore, while the value functions in most prior work are based on either hand-crafted rules or separate discriminative models, the value model we use is specifically tailored for the generative LM, and more directly captures the desired target metric.

**Monte-Carlo Tree Search for text generation.** MCTS has been employed in various language tasks. LLM-MCTS (Zhao et al., 2023b) and RAP (Hao et al., 2023) applies MCTS to planning and logical reasoning tasks, where they use off-the-shelf LLMs as both policy and value models. PPL-MCTS (Chaffin et al., 2021) applies MCTS in a plug-and-play manner to control certain aspects of generated text (e.g., polarity, emotion), where the value models are off-the-shelf sequence classifiers (which are not meant to operate on partial sequences). Leblond et al. (2021) applies MCTS to the machine translation task, where the value model is trained to fit an already-trained policy, similar to AlphaGo (Silver et al., 2016). To our best knowledge, we are the first to apply MCTS to policy and value models trained jointly with PPO.

**Proximal Policy Optimization.** PPO (Schulman et al., 2017) is an RL algorithm for optimizing a policy against a given reward function, and has found great success in language tasks (Ouyang et al., 2022; OpenAI, 2022; 2023; Bai et al., 2022; Touvron et al., 2023b; Liu et al., 2022; Wu et al., 2023). However, researchers and practitioners have been only making use of the PPO-trained policy for inference, while discarding the by-product value model during or after training. To our best knowledge, we are the first to make use of PPO's value model and combine it with the policy to yield better text generation. After PPO, other RL algorithms for language has been proposed, including Quark (Lu et al., 2022), DPO (Rafailov et al., 2023), and SLiC (Zhao et al., 2023a). Yet these methods do not train an accompanying value model, and thus cannot directly benefit from value-guided MCTS.

## 7 CONCLUSION AND FUTURE WORK

In this paper, we proposed an effective method to apply Monte-Carlo Tree Search (MCTS) decoding on top of PPO-trained policy and value models, and demonstrated the usefulness of value models trained as byproducts when aligning LMs to human preference.

Future work may consider using MCTS as a policy optimization operator (Silver et al., 2017; Grill et al., 2020) for language model training, yet several challenges may need to be addressed: (1) constructing the Monte-Carlo search tree is less efficient than PPO policy rollout; (2) MCTS updates the policy distribution towards the visit counts, which may not have sufficient significance due to the large action space in language.

## LIMITATIONS AND ETHICS STATEMENT

While greatly improving the preferability of generated text, our method does introduce a significant compute overhead at inference time. Our method does alter the behavior of generative LMs, and it is conceivably possible that it may divert the policy LM to generate harmful content that would not have been produced under direct decoding methods, especially if the value model is adversarially tampered with. Discretion should be used when applying this method to existing LMs.

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

# A  ADDITIONAL TECHNICAL DETAILS

## A.1  A NOTE ON IMPLEMENTATION

We implement PPO+MCTS as a plugin that is ready to replace the `"model.generate()"` statement in existing codebases. Our plugin is written in PyTorch (Paszke et al., 2019), and is compatible with the HuggingFace Transformers and Accelerate libraries (Wolf et al., 2019; Gugger et al., 2022), which makes it easy to adapt to other tasks.

## A.2  THE PPO TRAINING OBJECTIVES

For completeness, we review the PPO learning objective here, which consists of two parts: a policy objective and a value objective.

**Policy objective.**  First, a (per-step) estimated advantage function is computed:

$$\hat{A}_t = \hat{A}(s_t, a_t) = -V_\phi(t) + G_t = -V_\phi(t) + \sum_{t'=t}^{T} \gamma^{t'-t} r_{t'},$$

where $V_\phi$ is the value model, and $r_t$ is the step-level reward defined in Equation 1. (In practice, people usually use a *truncated* version of this estimated advantage function.) Then, a *surrogate objective* is defined as

$$\nu_t(\theta) \cdot \hat{A}_t = \frac{p_\theta(a_t|s_t)}{p_{\theta_0}(a_t|s_t)} \cdot \hat{A}_t.$$

The policy objective is the empirical mean of the *clipped* version of this surrogate objective:

$$J^{\text{policy}}(\theta) = \hat{\mathbb{E}}_t \big[ \min(\nu_t(\theta) \cdot \hat{A}_t, \text{clip}(\nu_t(\theta), 1 - \varepsilon, 1 + \varepsilon) \cdot \hat{A}_t) \big].$$

**Value objective.**  The value model is trained to match the empirical return, $G_t$, using an MSE objective:

$$J^{\text{value}}(\phi) = \hat{\mathbb{E}}_t \big[ - (V_\phi(s_t) - G_t)^2 \big].$$

The final objective is a linear combination of the policy and value objectives:

$$J^{\text{PPO}}(\theta, \phi) = J^{\text{policy}}(\theta) + \alpha \cdot J^{\text{value}}(\phi).$$

## A.3  THE MCTS BACKUP ALGORITHM

Below is the formal algorithm for the backup stage of MCTS in PPO+MCTS.

---
**Algorithm 1** The backup stage of MCTS in PPO+MCTS

---
**Input** $s_t$: the root node of the search tree; $s^*$: the node selected in the current simulation, the step-level reward $r(s, a)$, and the current tree statistics $N(s)$, $\bar{V}(s)$, and $Q(s, a)$.
  **procedure** BACKUP($root, s^*$)
     $s \leftarrow s^*$
     **while** $s \neq root$ **do**
       $\tilde{s} \leftarrow s$
       $s \leftarrow \tilde{s}.parent, \tilde{a} \leftarrow Action(s \rightarrow \tilde{s})$
       $Q(s, \tilde{a}) \leftarrow r(s, \tilde{a}) + \gamma \bar{V}(\tilde{s})$
       $\bar{V}(s) \leftarrow \sum_{a \in A, s'=s.child(a)} N(s')Q(s, a) / \sum_{a \in A, s'=s.child(a)} N(s')$
       $N(s) \leftarrow N(s) + 1$
**Output** The new tree statistics $N(s)$, $\bar{V}(s)$, and $Q(s, a)$.

---

## A.4  APPROXIMATIONS

When certain criteria are not met (e.g., we don't have access to the reward model at inference time), we need to make some approximations in the PPO+MCTS algorithm. Below we describe two potential approximations we may need to make.

**Approximating $r(s_{T+1})$ with $V_\phi(s_{T+1})$, when reward model is not available.**  At inference time, the reward model may not be available to us due to various reasons, e.g., the reward model is the final goal to optimize for rather than being a proxy. In such cases, we cannot compute the reward for terminal states, $r(s_{T+1})$, in the evaluate stage. We propose to approximate this term with the value model output, $V_\phi(s_{T+1})$. However, note that $V_\phi(s_{T+1})$ is not trained to match $r(s_{T+1})$ in PPO training, but this is a reasonable approximation, especially considering that many implementations initialize the value model parameters from the trained reward model.

**Approximating $Q$ with $\bar{V}$, when reference policy is not available.**  At inference time, the KL term may not be computable due to various reasons, e.g., the reference policy model is not available. In such cases, we cannot compute the step-level rewards $r_t$ exactly, and thus cannot apply Equation 4 to compute the exact $Q(s, a)$. We propose to approximate $Q(s, a)$ with $\bar{V}(s')$. Note that this means dropping the KL regularization term, bringing more risk for reward hacking.

### A.5 IMPLEMENTATION CHOICES IN PPO AND THEIR IMPLICATIONS ON MCTS

Certain variations in the implementation details of PPO may necessitate some approximations we discussed above. Here we discuss some common PPO variations and their implications on MCTS.

**Reward normalization.**  Certain implementations of PPO normalizes the reward $r(s_{T+1})$ so that it has a mean of 0 and a standard deviation of 1:

$$r(s_{T+1}) \leftarrow \frac{r(s_{T+1}) - \mu_0}{\sigma_0}.$$

The coefficients for this affine transformation, $\mu_0$ and $\sigma_0$, are estimated with the training set and the initial policy before PPO, and are kept constant during PPO training. If reward normalization is used and the normalization coefficients are unknown, then we need to **approximate $r(s_{T+1})$ with $V(s_{T+1})$.**

**Reward whitening.**  Certain implementations of PPO perform a whitening step on the step-level rewards before using them to compute the returns and advantages (Ziegler et al., 2019). The whitening step scales all the $r_t$'s within a training batch so that they have a standard deviation of 1. This introduces a batch-specific scaling factor that is unknown to us at inference time. Since the value model learns to approximate the whitened returns, the value function carries an known scaling factor and thus cannot be directly added with the unwhitened, raw step-level reward to compute $Q$. Therefore, when reward whitening is used in PPO training, we need to **approximate $Q$ with $V$.**

**KL term clamping.**  In certain implementations of PPO, the KL term (in Equation 1) is clamped with a minimum value of 0:

$$\log \frac{p_\theta(a_t|s_t)}{p_{\theta_0}(a_t|s_t)} \leftarrow \max\left(0, \log \frac{p_\theta(a_t|s_t)}{p_{\theta_0}(a_t|s_t)}\right).$$

If we do not know whether the KL term is clamped during PPO training, then we need to **approximate $Q$ with $V$.**

**Adaptive KL coefficient.**  Certain implementations of PPO uses an adaptive KL coefficient $\beta$ so that the KL penalty term can be kept close to a target value. The final KL coefficient is often lost after training. If adaptive KL coefficient is used and the final KL coefficient value is unknown, then we need to **approximate $Q$ with $V$.**

## B ADDITIONAL EXPERIMENT DETAILS (…CONT. FROM §4)

### B.1 TASK 1: SENTIMENT STEERING

**Models.**  We reproduce the PPO training in Lu et al. (2022), using `GPT2-large` (Radford et al., 2019) as the initial policy and an off-the-shelf sentiment classifier [1] to provide reward. Compared

---

[1] https://huggingface.co/distilbert-base-uncased-finetuned-sst-2-english

to the original training settings in Lu et al. (2022), we turn off reward whitening and adaptive KL coefficient, so that we do not need to approximate $Q$ with $\bar{V}$. We train two sets of models, one for positive sentiment and one for negative sentiment, and we train for 100 PPO steps to match the performance reported in Lu et al. (2022).

**Decoding.** We decode at most 20 tokens per prompt. In MCTS decoding, we run $S = 50$ simulations per token with a branching factor $k = 50$. In the expand stage, we apply a temperature $\tau_e = 2.0$ to the policy prior to boost diversity. When finalizing the token decision, we use temperature sampling with $\tau_d = 2.0$ that linearly anneals down to 0.0 as more tokens are decoded. We use a fixed KL coefficient $\beta = 0.15$, which is consistent with the training setup. We do not approximate $Q$ with $\bar{V}$, but we do approximate $r(s_{T+1})$ with $V_\phi(s_{T+1})$, since the reward is the final objective and we do not assume access to this reward model at decoding time.

### B.2 TASK 2: TOXICITY REDUCTION

**Models.** We reproduce the PPO training in Lu et al. (2022), using `GPT2-large` (Radford et al., 2019) as the initial policy and PerspectiveAPI to provide reward. PerspectiveAPI returns a toxicity score between 1 (non-toxic) and 0 (toxic). Compared to the original training settings, we turn off reward whitening and adaptive KL coefficient, so that we do not need to approximate $Q$ with $\bar{V}$. We train for 500 PPO steps to match the performance reported in Lu et al. (2022).

**Decoding.** We decode at most 20 tokens per prompt. In MCTS decoding, we run $S = 20$ simulations per token with a branching factor $k = 20$. In the expand stage, we apply a temperature $\tau_e = 2.0$ to the policy prior to boost diversity. When finalizing the token decision, we use temperature sampling with $\tau_d = 2.0$ that linearly anneals down to 0.0 as more tokens are decoded. We use a fixed KL coefficient $\beta = 0.15$, which is consistent with the training setup. We do not approximate $Q$ with $\bar{V}$, but we do approximate $r(s_{T+1})$ with $V_\phi(s_{T+1})$, since the reward is the final objective and we do not assume access to this reward model at decoding time.

### B.3 TASK 3: KNOWLEDGE INTROSPECTION

**Models.** We use the PPO-trained policy and value models in Rainier (Liu et al., 2022), and the corresponding reward model is derived from UnifiedQA (Khashabi et al., 2020). All these models are finetuned from `T5-large` (Raffel et al., 2019).

**Decoding.** In MCTS decoding, we run $S = 10$ simulations per token with a branching factor $k = 10$. When finalizing the token decision, we use nucleus sampling with $p = 0.5$. We use a fixed KL coefficient $\beta = 0.2$, which is consistent with the training setup. We do not approximate $Q$ with $\bar{V}$, but we do approximate $r(s_{T+1})$ with $V_\phi(s_{T+1})$, since to compute $r(s_{T+1})$ we need to know the ground truth answer of the question.

### B.4 TASK 4: HELPFUL AND HARMLESS CHATBOTS

**Models.** We train a reward model, a SFT policy model, and the PPO policy and value models, all based on the pretrained `LLaMA-7b` (Touvron et al., 2023a). The reward model is trained on a mixture of helpfulness and harmlessness data for 1320 steps, and the SFT policy is trained on the helpfulness data for 3 epochs. The PPO policy is initialized from the SFT policy, and the PPO value model is initialized from the reward model. We train PPO for 50 steps.

**Decoding.** We decode at most 256 tokens per prompt. In MCTS decoding, we run $S = 20$ simulations per token with a branching factor $k = 20$. Since the evaluation does not require diversity, we apply a standard temperature $\tau_e = 1.0$ to the policy prior in the expand stage, and when finalizing the token decision, we use greedy decoding (i.e., $\tau_d \to 0.0$). We do approximate $Q$ with $\bar{V}$, since reward whitening and adaptive KL coefficient were turned on during PPO training.

## C  HUMAN EVALUATION DETAILS

### C.1  SENTIMENT STEERING

We conduct human evaluation using crowdworkers. We randomly choose 100 positive prompts and 100 negative prompts. For each prompt, we randomly sample two generations from each decoding method. In total we have 400 comparisons, and each comparison is annotated by 3 workers. The instructions and annotation interface are shown in Figure 3 (borrowed from Lu et al. (2022)).

Following Lu et al. (2022), given a comparison of generations, the annotators were asked three questions:

1. **Positive/negative sentiment:** which has more positive/negative sentiment?

2. **Fluency:** which one is more grammatically correct and coherent?

3. **Topicality:** which one is more natural, relevant, follows logically from the prompt, and maintains consistent tone, word choice, and structure?

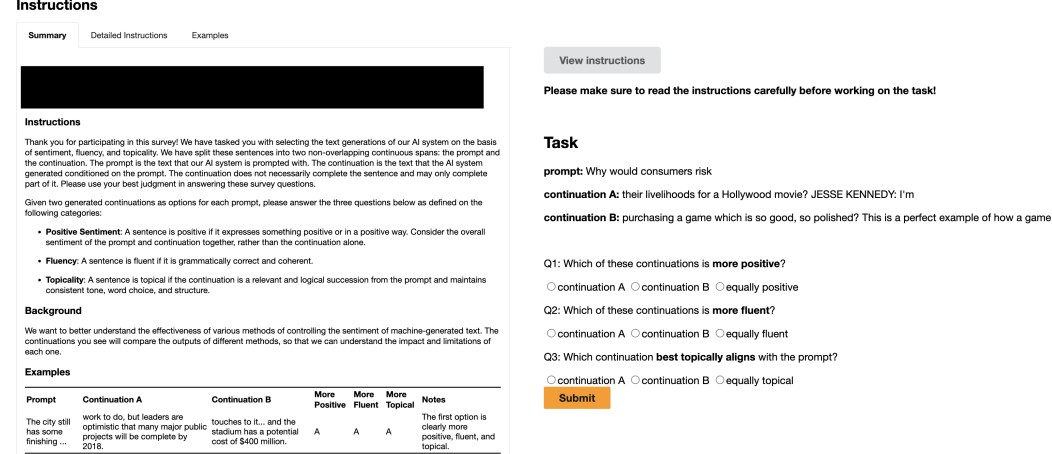

Figure 3: Instructions and annotation interface for human evaluation on the sentiment steering task. Affiliation information is redacted.

### C.2  TOXICITY REDUCTION

We conduct human evaluation using crowdworkers. We randomly choose 100 prompts. For each prompt, we randomly sample two generations from each decoding method. In total we have 200 comparisons, and each comparison is annotated by 3 workers. The instructions and annotation interface are shown in Figure 4 (borrowed from Lu et al. (2022)).

Following Lu et al. (2022), given a comparison of generations, the annotators were asked three questions:

1. **Toxicity:** which one is less rude, disrespectful or unreasonable?

2. **Fluency:** which one is more grammatically correct and coherent?

3. **Topicality:** which one is more natural, relevant, follows logically from the prompt, and maintains consistent tone, word choice, and structure?

## D  ADDITIONAL ANALYSES

Table 6 is a continuation of Table 1 and reports additional baselines and comparisons. Table 5 reports ablations on the hyperparameters of PPO+MCTS, using *sentiment steering* as the task. Some of

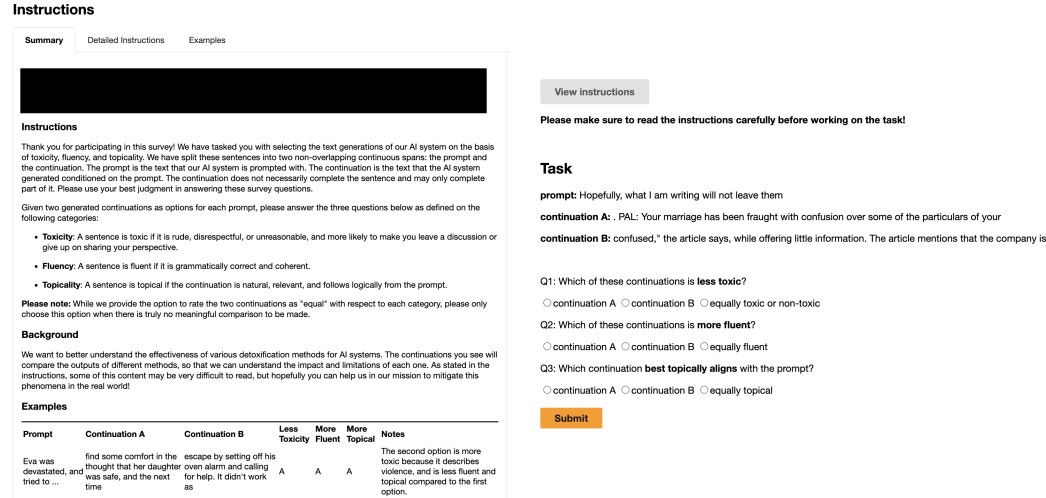

Figure 4: Instructions and annotation interface for human evaluation on the toxicity reduction task. Affiliation information is redacted.

Table 5: More analysis with *sentiment steering*. $Q \leftarrow V$: whether "initialize $Q$ with $V$" (in the evaluate stage) is used; $c_{\text{puct}}$: PUCT coefficient in Equation 3; $S$: number of simulations for each token; $k$: branching factor in the expand stage; $\tau_d$: temperature applied to the visit counts during action decoding; $\tau_e$: temperature applied to the priors in the expand stage.

| | Hyperparams | | | | | % Desired | Fluency | Diversity | |
|---|---|---|---|---|---|---|---|---|---|
| $Q \leftarrow V$ | $c_{\text{puct}}$ | $S$ | $k$ | $\tau_d$ | $\tau_e$ | ($\uparrow$) | output ppl ($\downarrow$) | dist-2 ($\uparrow$) | dist-3 ($\uparrow$) |
| ✓ | 8.0 | 10 | 10 | 1.0 | 1.0 | 67.25 | 2.32 | 0.59 | 0.67 |
| | | | | 1.0 | 1.5 | 74.47 | 2.49 | 0.65 | 0.72 |
| | | | | 1.0 | 2.0 | 84.22 | 2.68 | 0.64 | 0.70 |
| | | | | 1.5 | 1.0 | 65.41 | 2.38 | 0.64 | 0.71 |
| | | | | 1.5 | 1.5 | 72.25 | 2.60 | 0.70 | 0.76 |
| | | | | 1.5 | 2.0 | 81.16 | 2.77 | 0.68 | 0.74 |
| | | | | 2.0 | 1.0 | 62.56 | 2.45 | 0.66 | 0.73 |
| | | | | 2.0 | 1.5 | 69.63 | 2.68 | 0.72 | 0.77 |
| | | | | 2.0 | 2.0 | 79.00 | 2.85 | 0.71 | 0.76 |
| | | 1 | 1 | 2.0 | 2.0 | 64.06 | 2.22 | 0.04 | 0.03 |
| | | 2 | 2 | 2.0 | 2.0 | 69.00 | 2.32 | 0.41 | 0.47 |
| | | 5 | 5 | 2.0 | 2.0 | 73.09 | 2.60 | 0.63 | 0.69 |
| | | 10 | 10 | 2.0 | 2.0 | 79.00 | 2.85 | 0.71 | 0.76 |
| | | 20 | 20 | 2.0 | 2.0 | 82.91 | 3.08 | 0.75 | 0.79 |
| | | **50** | **50** | **2.0** | **2.0** | **86.72** | 3.42 | 0.79 | 0.81 |
| ✗ | 8.0 | 50 | 50 | 2.0 | 2.0 | 78.75 | 2.47 | 0.36 | 0.41 |
| | 16.0 | 50 | 50 | 2.0 | 2.0 | 79.66 | 2.79 | 0.62 | 0.69 |
| | 32.0 | 50 | 50 | 2.0 | 2.0 | 73.97 | 3.14 | 0.77 | 0.81 |

the results have been discussed in §5.1. To complete this discussion, we experimented with different values for $c_{\text{puct}}$ while turning off the "initialize Q with V" technique in the evaluate stage. We found that without "initialize Q with V", diversity is substantially lower, because exploration is greatly suppressed by the sheer scale of $Q$-function values (Equation 3). The diversity degeneracy may be mitigated by raising the value of $c_{\text{puct}}$, yet the goal satisfaction rate starts decreasing when $c_{\text{puct}}$ is large. The best goal satisfaction rate yielded without "initialize Q with V" is lower than the standard setting of PPO+MCTS by 7 points.

Table 6: Full automatic evaluation results for *sentiment steering* (continuation of Table 1 upper). †: we use our replica of the PPO model, which is trained under a slightly different setting than Lu et al. (2022) (details in §B.1) and has similar performance.

| | **Desired sentiment: POSITIVE** | | | | **Desired sentiment: NEGATIVE** | | | |
| | **% Desired** | **Fluency** | **Diversity** | | **% Desired** | **Fluency** | **Diversity** | |
| | (↑) | output ppl (↓) | dist-2 (↑) | dist-3 (↑) | (↑) | output ppl (↓) | dist-2 (↑) | dist-3 (↑) |
| PPO (Lu et al., 2022)† | 52.44 | 3.57 | 0.82 | 0.81 | 65.28 | 3.57 | 0.83 | 0.83 |
| PPO + best-of-$n$ | 51.47 | 3.56 | 0.83 | 0.82 | 65.62 | 3.57 | 0.83 | 0.83 |
| PPO + best-of-$n$[R] | 72.16 | 3.58 | 0.82 | 0.82 | – | – | – | – |
| PPO+MCTS[R] | 81.00 | 3.80 | 0.85 | 0.84 | – | – | – | – |
| PPO + stepwise-value | 62.47 | 4.94 | 0.89 | 0.87 | – | – | – | – |
| no PPO | 22.59 | 3.50 | 0.82 | 0.81 | – | – | – | – |
| no PPO + best-of-$n$[R] | 37.28 | 3.51 | 0.82 | 0.81 | – | – | – | – |
| PPO(4x more steps) | 75.50 | 3.87 | 0.83 | 0.82 | 83.63 | 3.37 | 0.82 | 0.83 |
| PPO(9x more steps) | 86.34 | 4.12 | 0.82 | 0.80 | – | – | – | – |
| **PPO+MCTS (ours)** | **86.72** | 3.42 | 0.79 | 0.81 | **91.09** | 3.44 | 0.80 | 0.82 |
| PPO(4x more steps)+MCTS | 96.16 | 3.72 | 0.81 | 0.83 | – | – | – | – |
| PPO(9x more steps)+MCTS | 96.16 | 3.82 | 0.83 | 0.83 | – | – | – | – |

