# OpenReview forum: "Making PPO even better: Value-Guided Monte-Carlo Tree Search decoding"
_ICLR.cc/2024/Conference — Submitted to ICLR 2024_

### Official Review · Reviewer_mGKi · 2023-10-27

**Soundness:** 2 fair
**Presentation:** 3 good
**Contribution:** 2 fair
**Rating:** 5
**Confidence:** 4

**Summary:**

This paper introduces a novel text decoding algorithm named "PPO-MCTS" which integrates Monte Carlo Tree Search (MCTS) on top of Proximal Policy Optimization (PPO). The study evaluates the effectiveness of this approach by comparing it to other PPO-based algorithms in four text generation tasks: sentiment steering, toxicity reduction, knowledge introspection, and developing helpful and harmless chatbots. The results indicate that the proposed method outperforms other PPO-based algorithms in terms of performance.

**Strengths:**

The paper is well-written and provides clear motivation. The proposed method PPO-MCTS, which combines the trained PPO language model and the MCTS method, is straightforward and effective.

**Weaknesses:**

One of the major weaknesses is the inference time of PPO-MCTS. As the authors said, "... PPO-MCTS is 2S times slower due to the tree construction, where S is the number of simulations ...". It is questionable that this method can be used in practice.

In addition, the current experiments are not quite convincing, as explained below.
* In section B.1, "Compared to the original training settings in Lu et al. (2022), we turn off reward whitening and adaptive KL coefficient, …", and in section B.2 "Compared to the original training settings, we turn off reward whitening and adaptive KL coefficient, …". This shows that the authors use different training settings for the baseline model, which leads to different performances. In Table 3, the PPO (Liu et al, 2022) result is evaluated by the author. However, the performance (average QA accuracy) is lower than the reference (58.69 vs. 60.36). The performance in (Liu et al, 2022) is even higher than PPO-MCTS.
* In Table 1, "PPO (4x more steps)" outperforms "PPO" (75.50 vs. 52.44), and the result is getting close to PPO-MCTS. Does "PPO (4x more steps)" converge? Why not try more steps for PPO, e.g. PPO (8x more steps)? In addition, how is the performance of PPO(4x more steps)-MCTS?
* Since the PPO-MCTS is much slower than PPO, it would be better to evaluate PPO-MCTS with 1-50 simulations instead of only 10, 20, and 50 simulations. This result can provide readers with a tradeoff between performance and evaluating time.

**Questions:**

* The term "usefulness" in Table 3 is not defined.
* How to apply the temperature $\tau_e$ to the priors $p_\theta(a | s)$ in "How to get diversity out of MCTS?" in section 5.1?
* In Table 3, the PPO (Liu et al, 2022) result is evaluated by the author. However, the performance (average QA accuracy) is lower than the reference (58.69 vs. 60.36). The performance in (Liu et al, 2022) is even higher than PPO-MCTS.
* What does $r$ represent in equation (4)? Is that r(s)?
* Does "PPO (4x more steps)" converge? Why not try more steps for PPO, e.g. PPO (8x more steps)?
* How is the performance of PPO(4x more steps)-MCTS?
* To balance the tradeoff between performance and evaluating time, it would be better to evaluate PPO-MCTS with 1-50 simulations instead of only 10, 20, and 50 simulations.
* In the first sentence in section 3, "Applying MCTS decoding on top of PPO-trained models *allow* ..." -> "Applying MCTS decoding on top of PPO-trained models *allows* ..."

---

> ### Author Response · Authors · 2023-11-15
>
> Thanks for the insightful feedback! We really appreciate your recognition of our paper’s clarity and the effectiveness of our method.
>
> Here is our response to your concerns: (we have updated our paper submission to reflect the appropriate changes, changes are rendered in blue font color)
> 1. **Inference-time efficiency:** The significant inference-time overhead is indeed a weakness of our method, which we have acknowledged in the “Limitations” section. We argue that PPO-MCTS may be found useful in several scenarios:
>     * When the quality of generated text is critical, PPO-MCTS offers a way to improve text quality out of existing trained models, at some expense of additional inference-time compute;
>     * Our method shows the promise of using MCTS to explore and rollout better trajectories during PPO training (improving upon the commonly-used temperature-sampling rollout), which we briefly mentioned in the “Conclusions and Future Work” section. This has the potential to improve PPO training itself, and will not incur additional inference-time latency (assuming not using MCTS for inference). To achieve this, our proposed PPO-MCTS decoding method would be an indispensable component.
> 1. **Turning off reward whitening:** We acknowledge that our PPO training settings are slightly different from the original in Lu et al. (2022). However, the baseline we report and compare with is the PPO model trained with our settings, so our comparisons are fair. We added a note in Tables 1&2 to clarify this.
> 1. **Higher baseline for knowledge introspection:** Our experimental results on the PPO baseline are slightly lower than Liu et al. (2022), possibly due to discrepancy in compute environments and random variation. For consistency, we compared PPO-MCTS with the baseline results produced by our experiments.
> 1. **PPO with even more steps:** Thanks for your suggestion, we added more experimental results to Table 6 (rows “PPO(9x more steps)”, “PPO(4x more steps)+MCTS”, and “PPO(9x more steps)+MCTS”). PPO converged at “9x more steps”, and yet its goal satisfaction rate (86.34) is still lower than PPO+MCTS (86.72). MCTS is able to improve the goal satisfaction rate by at least 10 points across all these PPO models.
> 1. **MCTS with fewer simulations:** We added more experimental results to Table 5, with the number of simulations $S=\{1,2,5\}$ (in addition to existing results where $S=\{10,20,50\}$). (Note that $S=1$ reduces MCTS to greedy decoding, and thus has very low diversity metrics.) We do observe that the goal satisfaction rate steadily increases as $S$ increases.
> 1. **Applying temperatures to priors:** To get diversity, we can apply a temperature $\tau_e$ when computing the prior $p(a|s)$ by following: Let $l(a|s)$ be the pre-softmax logits of $p(a|s)$. Normally you would do $p(a|s) = \exp{(l(a|s))} / \sum_{a’} \exp{(l(a’|s))}$. When applying a temperature, we do $p(a|s) = \exp{(l(a|s) / \tau_e)} / \sum_{a’} \exp{(l(a’|s) / \tau_e)}$. When using a $\tau_e > 1.0$, the distribution becomes less spiky.
> 1. **The “r” in Eq.4:** We apologize for the confusion. We improved the notation in Equations 4-6, where the previous $r$ is actually $r(s, \tilde{a})$, the step-level reward for taking a hypothetical action $\tilde{a}$ at state $s$. Since this is an unrealized, hypothetical decoding state living deep down in the search tree, we can’t use $r_t$ to denote this reward, but the definition is the same as $r_t$ in Eq.1. We also added an algorithm (Algorithm 1, in Appendix A.3) to clarify the backup process of MCTS.

---

> > ### Comment · Reviewer_mGKi · 2023-11-23
> >
> > Thank you for addressing my questions. However, the time of PPO-MCTS is still a big issue (2S times slower). Although the PPO-MCTS may offer a way to improve text quality, it will be more convincing to provide additional experiments. Overall, I tend to keep my score.

---

### Official Review · Reviewer_Svcj · 2023-10-31

**Soundness:** 2 fair
**Presentation:** 2 fair
**Contribution:** 2 fair
**Rating:** 3
**Confidence:** 5

**Summary:**

The paper proposes a combination of PPO and MCTS for natural language text generation. The MCTS algorithm uses both the policy and value output of the PPO network, and has some modifications to a standard MCTS algorithm. The empirical analysis show that the proposed algorithm performs better than some PPO-based baselines on four benchmarks.

**Strengths:**

The empirical results indicate that the proposed algorithm is competitive for the tasks considered.

**Weaknesses:**

The description of the MCTS variant is somewhat difficult to understand.

**Questions:**

Most of the important results with the MCTS algorithm have been achieved in combination with off-policy RL algorithms. In those instances both the policy and value outputs are used by MCTS. When PPO is used standalone, the value output is not needed anymore, but as noted in the paper, it is useful when PPO is combined with MCTS. However, in the view of the well known results with off-policy RL algorithms, this is hardly surprising.

In the presentation of the new MCTS, there are some alteration highlighted. I am not sure of the novelty of most of these changes.

- "replacing V with Q": even the original UCT paper defined the problem as MDP, with non-terminal rewards included. Using Q(s,a) is much more standard than V(s), which has an after-state flavor. Using the KL penalty term as (or in the) reward seems a bit odd. It is steering the search toward the original policy, which may not be a bed thing, but it is already done by the use of prior in equation (3). The KL term is not vanishing with additional search, which means that the MCTS algorithm does not converge asymptotically to an optimal action.

- "initializing Q with V": this is probably the standard way of using value functions in MCTS

- "Forbidding node expansion after terminal states": this is just the normal thing to do. Probably, there is no MCTS implementation that would expand the tree beyond a terminal state.

The backup rule, as defined looks somewhat strange. The update (5) leads to NaN if any of of the child nodes (s') is not yet sampled (N(s')=0). The usual backup rule is to update it with the return from the current trace. Maybe it is happening here as well, but it is awkwardly written. It would be useful to describe the algorithm with a pseudocode, so that similar issues can be clarified better.

---

> ### Author Response · Authors · 2023-11-15
>
> Thanks for your valuable feedback! We really appreciate your recognition of the competitiveness of our approach.
>
> Below is our response to your concerns: (we have updated our paper submission to reflect the appropriate changes, changes are rendered in blue font color)
> 1. **Prior success of MCTS on off-policy RL:** Thanks for noting that prior work has reported success of MCTS in combination with off-policy RL algorithms. We’d like to note that PPO is an on-policy RL algorithm, and therefore our findings may be seen as an extension to existing knowledge of what type of models MCTS can be useful with.
> 1. **Novelty:**
>     * **Replacing V with Q:** Thanks for pointing out the original source of this idea. Other reviewers (GmAu and ZQMA) also agree with this. We added a citation to Kocsis and Szepesvari (2006), and removed this point from our novelty claim.
>     * **The KL term:** We appreciate your perspective, but we respectfully hold a different viewpoint on this. The goal of search is *not* to make the KL term vanish – the KL term is conventionally included in the PPO training objective, and in MCTS it is a part of the value/Q functions that guides the search. Also, the KL term steers decoding towards the reference policy, while the prior in Eq 3 makes the decision informed by the PPO policy (which is regularized by the KL term to stay close to the reference policy during PPO training).
>     * **Initializing Q with V:** Based on our investigation of several MCTS implementations as listed below,  this technique is not adopted, so we concluded that it is *not* standard for MCTS. If you can share evidence that shows otherwise, we’d greatly appreciate it.
> [[1](https://github.com/google-deepmind/open_spiel/blob/6e46a2d5746250cddb5bcc3124c73415f0c18732/open_spiel/algorithms/mcts.cc#L366-L428), [2](https://github.com/NohTow/PPL-MCTS/blob/ed588ca8f8abf4dd113f57d6c95c804ef9ca961e/mcts_rollout.py#L486-L521), [3](https://github.com/ImparaAI/monte-carlo-tree-search/blob/469ef42a1dc49aa1dbe4e6fd16bef05c200267d7/montecarlo/montecarlo.py)]
>     * **Forbidding exploration after terminal states:** Thanks for pointing out that this is standard practice. Reviewer (GmAu) also agrees with this. We removed this point from our novelty claim.
> 1. **Backup algorithm:** We apologize for the confusion. We improved the notation in Equations 4-6, and added an algorithm (Algorithm 1, in Appendix A.3) to clarify the backup process of MCTS. To answer your question, a NaN result cannot happen, because to get a zero denominator you need $N(s’) = 0$ for all $s’$ (where $s’$ is a child node of state $s$, where the child edge (a.k.a., the action taken) is $a$). At least one of the $N(s’)$ must be non-zero, because $s$ is an intermediate node and the “expand” stage of the current simulation round has explored one of its subtrees.
> 1. **Clarity:** Hopefully our response point #3 above addresses your comment that our description of MCTS “is somewhat difficult to understand”, since aside from the modifications we carefully described, the MCTS we use is standard as in prior literature. If there’s anything else we can make it easier to follow, please don’t hesitate to let us know ;)

---

> > ### Comment · Reviewer_Svcj · 2023-11-22
> >
> > I appreciate the effort of the authors to improve the paper, but I still do not have the feeling that it is close to the level of acceptance.
> >
> > The KL term in PPO is used to make the policy search process more stable. It is not a term that one would want to optimize in general. There can be an argument for keeping the search based policy close to the PPO (prior) policy in the beginning of the search phase, but I do not see why one would want to keep it close to the prior after sufficiently large search.

---

### Official Review · Reviewer_GmAu · 2023-11-01

**Soundness:** 2 fair
**Presentation:** 1 poor
**Contribution:** 3 good
**Rating:** 3
**Confidence:** 4

**Summary:**

The submission investigates the use of MCTS for LLM decoding, using the value network from RLHF as MCTS's evaluator.

**Strengths:**

The submission's main idea is obvious (which is not a bad thing!) but untried insofar as I am aware. It pertains to an important problem, and so has a high potential for significance.

**Weaknesses:**

My first criticism regards the name of the submission and method. I'm not totally in agreement that the submission is about "Making PPO Better". The submission is about using MCTS to better align the model's output with human feedback. That the value function comes from PPO in particular seems not very material to the actual contribution of the submission. I also find the name PPO-MCTS somewhat misleading. To me, this evokes a method that combines PPO and MCTS in the spirit of PG Search (https://arxiv.org/abs/1904.03646) or RL Search (https://openreview.net/forum?id=D0xGh031I9m), which is not what the submission intends to convey. It would be more clear to label the approach PPO+MCTS.

My second criticism regards the submission's claims of novelty regarding MCTS, which I repeat below:

> 1. Replacing the edge values V with the Q-function to ensure consistency with PPO training.

I found the description of (1) given in the introduction somewhat unclear, but the description is clarified under equation (3), where the text states:

```
Note that this is different from the standard MCTS algorithm, where V(s′) is directly used in
place of Q(s, a) in Equation 3. We refer to this change as replacing V with Q. We made this
change because, in the standard MCTS setup, the reward is given only to the terminal step (i.e.,
intermediate steps are assumed a zero reward), and there is no discounting horizon (γ), and thus
the return is always equal to the final reward. However, in PPO, intermediate steps are penalized
with a KL term (Equation 1), and there is a discounting factor γ when computing the return
(Equation 2). We use the version in Equation 3 to faithfully capture this regularization term.
```
This is not novel -- it was actually the original formulation of MCTS. See *Bandit based Monte-Carlo Planning* (Kocsis and Szepesvari, 2006), which I notice the submission neglects to cite.

> 2. Initializing the Q of children actions from the V of their parent node to encourage exploration.

I'm not sure I understand why this is important. The submission states
```
We made this change because with PPO models, the Q can have rather
large scales (in the order of 10s), due to reasons that will be explained in §A.4. During early
experiments, we found that this can severely suppress exploration in the tree search, making it
degenerate to greedy decoding.
```
This explanation doesn't make sense. The scale of Q should not matter. As far as argmaxing equation (3), any increase in the scale of Q can be mitigated by an increase in the scale of c_puct.

> 3. Forbidding exploration after terminal states.

```
Forbidding node expansion after terminal states. Text generation usually stops at a terminal
token, [EOS]. When the action is [EOS], the child node is called a terminal node. Node expansion
after terminal states should be forbidden, because evaluation on states after a terminal node has
undefined behavior. To maintain proper visit counts up to the terminal node, when encountering a
terminal node in the select stage, we should not stop the simulation early, but jump directly to the
backup stage
```
This is not novel -- it's standard practice. See open source MCTS implementation: https://github.com/google-deepmind/open_spiel/blob/master/open_spiel/algorithms/mcts.h#L187-L191

---

My third criticism is structural. The submission's method section mostly just explains MCTS. But MCTS is not new methodology from the paper -- it's background material. The submission should describe MCTS as background material in the preliminaries section. The methodology section should just explain how the submission applies MCTS in its specific context. It's ok if the methodology section is short.

---

I also had this comment.

> Following Chaffin et al. (2021), we do not do Monte-Carlo rollout due to efficiency considerations.

This isn't an appropriate citation for not doing Monte Carlo rollouts for efficiency considerations. Using a value function instead of Monte Carlo rollouts is common practice going back at least to AlphaGo Zero.

**Questions:**

> Think of the things where a response from the author can change your opinion

I think the submission requires substantial re-structuring and re-characterization of contributions to be suitable for publication. I would possibly  change my opinion if the issues above were addressed.

---

> ### Author Response · Authors · 2023-11-15
>
> Thanks for the insightful feedback! We appreciate your recognition of the potential significance of our proposed method.
>
> Here is our response to your concerns: (we have updated our paper submission to reflect the appropriate changes, changes are rendered in blue font color)
> 1. **Title and method name:** We agree that our proposed method is an inference-time algorithm that complements PPO, rather than improving PPO itself. We will change our title to **Generating more preferable text with Value-Guided Monte-Carlo Tree Search decoding**. We will also change the method name to **PPO+MCTS** to better reflect that we are combining two existing approaches.
> 1. **Novelty:**
>     * **Replacing V with Q:** Thanks for pointing out the original source of this idea. Other reviewers (Svcj and ZQMA) also agree with this. We added a citation to Kocsis and Szepesvari (2006), and removed this point from our novelty claim.
>     * **The scale of Q and c_puct:** The unknown scaling factor in Q cannot be mitigated by tuning $c_\text{puct}$, because the unknown scaling factor is instance-specific (Appendix A.5). Empirically, we added experiments with different values of $c_\text{puct}$ while turning off the “initialize Q with V” technique: while increasing $c_\text{puct}$ does mitigate the exploration degeneracy, it causes degradation of task performance. For details please see additions to Table 5 (Appendix D).
>     * **Forbidding exploration after terminal states:** Thanks for pointing out that this is standard practice. Reviewer (Svcj) also agrees with this. We removed this point from our novelty claim.
> 1. **Structural:** Thanks for the suggestion. We believe it is better to keep the introduction of MCTS algorithm in the Method section for sake of fluency in narration, because there are several parts tailored to PPO models and language generation tasks.
> 1. **Using value function in place of Monte-Carlo rollout:** Indeed we can confirm that AlphaGo-Zero already uses this technique, and we added a citation there (in addition to Chaffin et al. (2021)).

---

> > ### Comment · Reviewer_GmAu · 2023-11-20
> > **Reviewer Response**
> >
> > > Title and method name
> >
> > Thanks for your amenability to constructive criticism here.
> >
> > > The scale of Q and c_puct
> >
> > I don't quite understand what the authors mean here. I read Appendix A.5 and I do not see any discussion on anything "instance specific". The argmax of equation (3) is invariant under positive multiplication, so it cannot be necessary simply because "Q can have rather large scales (in the order of 10s)" as explained in the text. I guess maybe the authors are suggesting that value variance is high inter-instance but low intra-instance? If that's case, isn't the natural thing to do to use advantages? Either way, I feel that the text as is doesn't clearly communicate the motivation for this change.
> >
> > > We believe it is better to keep the introduction of MCTS algorithm in the Method section for sake of fluency in narration, because there are several parts tailored to PPO models and language generation tasks.
> >
> > I actually think this is exactly why it would be better to extricate MCTS from the Method section. That way, the method section can really focus on the changes that were made to tailor to PPO and language generation tasks.
> >
> > >  Indeed we can confirm that AlphaGo-Zero already uses this technique, and we added a citation there (in addition to Chaffin et al. (2021)).
> >
> > This isn't a very significant point, but why attribute a paper that uses a natural approach four years after it was first done? I haven't read Chaffin et al. (2021) -- is there some reason this paper is more relevant to this technique than the many other papers that have also used it since 2017?

---

> > > ### Author Response · Authors · 2023-11-22
> > >
> > > Dear reviewer, Thanks for your feedback.
> > >
> > > >> The scale of Q and c_puct
> > >
> > > > I don't quite understand what the authors mean here. I read Appendix A.5 and I do not see any discussion on anything "instance specific". The argmax of equation (3) is invariant under positive multiplication, so it cannot be necessary simply because "Q can have rather large scales (in the order of 10s)" as explained in the text. I guess maybe the authors are suggesting that value variance is high inter-instance but low intra-instance? If that's case, isn't the natural thing to do to use advantages? Either way, I feel that the text as is doesn't clearly communicate the motivation for this change.
> > >
> > > Apologies for the confusion. As we discussed in Appendix A.5, Reward whitening "[...] introduces a batch-specific scaling factor that is unknown to us at inference time. Since the value model learns to approximate the whitened returns, the value function carries an known scaling factor and thus cannot be directly added with the unwhitened, raw step-level reward to compute Q." The batch-specific scaling factor in value function V cascades to an instance-specific scaling factor of Q, and thus $Q(s, a)$ (or $V(s)$, if we do the "approximate Q with V" technique) carries an implicit, unknown scale that is specific to the instance $(s, a)$.

---

### Official Review · Reviewer_ZQMA · 2023-11-10

**Soundness:** 3 good
**Presentation:** 2 fair
**Contribution:** 2 fair
**Rating:** 5
**Confidence:** 3

**Summary:**

This paper proposes to use the value function for inference-time decoding after RL training of a language model. Specifically, they propose to use MCTS with the value model guiding the search. Experiments are done on four tasks: sentiment steering with OpenWebText and reducing toxicity with RealToxicityPrompts (both initially from Quark by Lu et al, 2020), knowledge introspection similar to Rainier (Liu et al 2022) and creating a chatbot with Anthropic's HH dataset. The authors find that PPO + MCTS decoding with a value model outperforms just PPO, PPO + decoding n samples and re-ranking, as well as a simpler search algorithm that leverages the value model (stepwise-value).

**Strengths:**

The idea is relatively straightforward and so it is good to see experiments on so many benchmarks, especially covering the performance / KL tradeoff with metrics like fluency and diversity.

The experiments do well to show that the value model is a good choice for search, instead of a reward model, and that after PPO, MCTS seems to be better than best-of-n at decoding-time and a simpler greedy search using the value model.

It is also commendable that the authors write their code as a plugin for generation in the popular huggingface library. The authors also give great motivation and reasoning for their implementation choices for PPO and implication on MCTS in the appendix

**Weaknesses:**

The main weakness of the paper is the missing comparison axis of efficiency in the paper and the baselines it therefore decides to evaluate against. The fundamental issue is that RLHF is only used because it is a more efficient alternative to best-of-n decoding with a reward model. As shown in [Gao et al (2022)](http://arxiv.org/abs/2210.10760) as well as [Stiennon et al (2020)](http://arxiv.org/abs/2009.01325) just decoding $n$ samples from a model and choosing the best one using a reward model ("best-of-n") is as performant as RLHF and closer in KL to the original model. The reason this method isn't used is because it can require incredibly high $n$ and be computationally infeasible at scale. This paper proposes to do a computationally expensive method after training without really tackling the issue of computational cost. At minimum, the authors need to show how their method performs across different computation scales (values of $S$) and compare it to PPO, PPO+best-of-n, as well as best-of-n without PPO. The goal should be to give practitioners an idea of the computation/performance tradeoff of this method. The only results on a large-scale task, HH-RLHF, don't seem to be very strong which casts some doubt on the efficacy.

### evaluation and baselines

given the relative simplicity of the idea, I think there need to be more baselines that are convincing that 1. the value model is necessary and 2. that MCTS is the best way to use it

For 1. the existing experiment doing MCTS with a reward model is a good start but is not fully convincing
- best-of-n (with/wihout) PPO using the reward model

For 2. beam search is a much more popular decoding method
- beam search
- beam search using the value model

These seem a bare minimum to me, especially when there is a large literature on language model decoding. For comparison, [Leblond et al (2021)](https://arxiv.org/pdf/2104.05336.pdf) which the authors note is very related to their own work compared to all of the above as well as greedy decoding and MCTS with rollouts

### minor issues

the authors seem to imply that they have a novelty in MCTS by using a Q-value instead of a Value but if I'm not mistaken, this formulation is used by Leblond et al (2021)

one reasons given for the trick is also not applicable: though PPO does technically use a discount factor, it is generally set to 1 for LM decoding with a maximum length output as is done in RLHF

using only the training reward to measure performance of RLHF in the HH task is insufficient, it should be accompanied by a metric of KL from the initial model

despite having four tasks, only one of these HH, is related to the general tasks that RLHF is used for nowadays and there is a single evaluation of reward and a single note on relative performance with human comparisons. This is not sufficient

### stylistic

In general, the paper spends too much time on introduction, explaining existing ideas (e.g MCTS) and notation. For example, Paragraph 4 describing Figure 1 in detail is unnecessary and should really be captured by the caption.

Section 2.3 does not describe PPO but actor-critic methods in general, this is perfectly fine but should not be titled PPO.

**Questions:**

The main issues preventing me from recommending acceptance is evaluations that take into account efficiency and the baselines showing that both the value model and MCTS are necessary. I would recommend acceptance if the authors can demonstrate
- in which computational regimes their method outperforms best-of-n with/without PPO using the reward model
- those + beam search baselines on at least one benchmark (e.g. toxicity reduction would be sufficient)


### clarification

Why do you use nucleus sampling in HH
- This seems like an important choice and previous work seems to be just using regular sampling or greedy decoding

What exactly is PPO+best-of-n?
- Are you decoding $n$ samples and then choosing the best sample according to a reward model?
- If so, why are you using the value model instead of the reward model?

Why are your Fluency numbers so different from the original results in Quark?

Why do you remove the adaptive KL coefficient from experiments? It has been an important part of PPO performance and as long as you keep track of the final value, you shouldn't need to approximate Q with V, correct?

---

> ### Author Response · Authors · 2023-11-15
>
> Thanks for your detailed and thoughtful feedback! We really appreciate your recognition of the diversity of tasks and comprehensiveness in our experiments.
>
> Here is our response to your concerns: (we have updated our paper submission to reflect the appropriate changes, changes are rendered in blue font color)
> 1. **Efficiency as evaluation axis:** To investigate the efficiency-performance tradeoff, we added more experimental results to Table 5, and now it reports performance with the number of simulations $S=\{1,2,5,10,20,50\}$, representing increasing amounts of compute. We do observe that the goal satisfaction rate steadily increases with more inference-time compute.
> 1. **More experiments on best-of-n (with reward model):** Thanks for your suggestion, we added more experimental results to Table 6 (rows “PPO + best-of-$n$[R]” and “no PPO + best-of-$n$[R]”). PPO + best-of-$n$[R] has much lower performance than our PPO-MCTS (72% vs 87%). From these results, we can see that best-of-$n$ is not as good as MCTS, and thus MCTS is needed.
> 1. **Comparison with beam search:** Since diversity is an integral part of the *sentiment steering* task, beam search is not applicable there. We reported a baseline method similar to beam search, referred to as “stepwise-value” in Table 6 and also the 3rd paragraph of Sec 5.1. This method evaluates candidate tokens using the value function and keeps more promising tokens (as would beam search do), and supports diversity by sampling from the (softmax of) value function. As reported in Table 6, MCTS is substantially better than this stepwise decoding method.
> 1. **Novelty of “Replacing V with Q”:** Thanks for letting us know that this was adopted by Leblond et al. As other reviewers (GmAu and Svcj) pointed out, this technique was also adopted in the original formulation of MCTS. Therefore, we removed this point from our novelty claim.
> 1. **Q-function and discounting factor:** We acknowledge that most RLHF implementations set the discounting factor $\gamma = 1$. However, our reason for using the Q-function is still valid, because it captures the step-level reward $r(s, a)$ (Eq.4) which acts as a regularization term.
> 1. **KL in the HH-RLHF task:** Due to an accidental deletion of model checkpoints, we cannot retrieve the KL numbers on this task, and we apologize for this. In Table 4, we added the perplexity of generated text against the PPO reference policy (which was in the original experiment results we recorded). PPO-MCTS has similar perplexity as vanilla decoding from the PPO policy, which indicates that our method maintains good fluency and regularization on this task.
> 1. **Nucleus sampling in HH-RLHF:** We use nucleus sampling for the HH-RLHF task because, we experimentally found that using greedy decoding results in very verbose and repetitive outputs and has slightly lower reward, and temperature sampling can be unrobost due to the long-tail distribution issue (Holtzman et al. 2020).
> 1. **PPO+best-of-n, value model:** Yes, it means decoding $n$ samples and then choosing the best sample according to the PPO value model. We use the value model instead of reward model because, in some practical use cases, the reward model may be unavailable (e.g., in the toxicity reduction task, maybe PerspectiveAPI is offline or shut down). PPO-MCTS does not rely on the reward model, when using approximations (Appendix A.4).
> 1. **Fluency numbers:** Our fluency numbers look different than Quark’s because Quark used an *incorrect* perplexity formula (confirmed via their [code](https://github.com/GXimingLu/Quark/blob/30c9a0fe022b7d006b826abec2eb5d3ac288e637/main.py#L322) and by Quark’s author). Their PPL is the inverse probability of the *full generated sequence*, whereby the correct PPL should have a length-normalization factor. Also, single-digit PPLs on text generated by an LM (as in our results) are more plausible than double-digit numbers.
> 1. **Removing the adaptive KL:** We removed the adaptive KL coefficient because many recent PPO implementations have removed it by default (ref: [AlpacaFarm](https://github.com/tatsu-lab/alpaca_farm/blob/f8bfbc28c2d10e0fb24b8fc8bd7b1fc03654eef7/src/alpaca_farm/rl/ppo_utils.py#L88), [Fine-Grained RLHF](https://github.com/allenai/FineGrainedRLHF/blob/3ce87b80f999f61adfb440df12e14e1771ca4eef/tasks/qa_feedback/training/fine_grained_config.yml#L41), [Rainier](https://github.com/liujch1998/rainier/blob/051a339a5b05df316b1d0d51e4640c6ebed3bd77/rainier/args.py#L42-L43)), likely because it does not hurt performance in language tasks. It is definitely viable to keep track of the final value of the adaptive KL coefficient and use it for MCTS decoding; we kept the KL coef constant for simplicity.
>
> If there’s any additional experiments we can add to better support our proposed method, please don’t hesitate to let us know. We look forward to your further review, and happy to discuss in more depth.

---

> > ### Author Response · Authors · 2023-11-15
> >
> > One more response to your concerns --
> >
> > 11. **Stylistic:** The caption of Figure 1 and the paragraph 4 of the intro are somewhat complementary, and we kept them for completeness of the narration. We titled Section 2.3 “PPO” because our method is experimentally validated on PPO rather than A2C methods broadly, and we kept this title for precision of our scope; also, the discussion in Appendix A.2, which is a continuation of Section 2.3, is specific for PPO.

---

> > > ### Comment · Reviewer_ZQMA · 2023-11-17
> > > **Accidental De-anonymization**
> > >
> > > Hello authors, I will respond to your rebuttal soon but in the meantime, you've accidentally de-anonymized your submission in the latest paper upload. Please fix this ASAP

---

> > > > ### Author Response · Authors · 2023-11-17
> > > >
> > > > My deepest apologies, we have fixed the anonymization issue and re-uploaded the draft. Thanks for bringing this to our attention.

---

### Meta-Review · Area_Chair_1fND · 2023-12-02

**Metareview:**

Summary & Strenghts: The paper proposes to combine MCTS decoding with a PPO-trained policy, where the key idea is to utilize the value network from PPO to guide the search process. PPO + MCTS improves text generation in terms of task-specific rewards as well as human evals across four diverse tasks relative to PPO. Using PPO's value function for MCTS is a straightforward idea and potentially useful given additional compute at inference time. The authors also release code which would make adoption easier.

Weaknesses and what's missing:

- Inadequate Comparison to Baselines / Compute Scalability:

    - The paper proposes a straightforward application of MCTS with PPO that requires additional compute at inference time. As such, the efficacy of this method should be compared with other baseline approaches with the *same amount of inference compute*, in particular using (1) best-of-N using the reward model both with and without PPO and (2) beam search with and without the value model.

    - As pointed out by reviewers and acknowledged by authors, PPO + MCTS is more computationally intensive than the standard PPO, raising doubts about its practicality in real-world applications, where efficiency is often as crucial as effectiveness. How does PPO + MCTS scale with additional compute relative to the baseline inference-time decoding approaches?

(The authors added comparison to training PPO longer compared to using PPO + MCTS during the discussion period,  which I believe is a good addition but does not answer the above question.)

- Discrepancy with prior work and overclaims: Some claims of novelty regarding specific design choices in MCTS were challenged by multiple reviewers, as similar choices have been documented in prior work. Moreover, the paper deviates from the original training settings used in baseline models, leading to performance disparities and questioning the validity of the comparison results (which was partly addressed during discussion).

**Justification For Why Not Higher Score:**

There was a good discussion between the authors and reviewers and the authors addressed several concerns. Nevertheless, some major concerns about compute scalability and comparison to other inference-time decoding methods remain unaddressed, which are critical given the limited novelty and simplicity of the idea.  Overall, I agree with the reviewers that this paper has potential but requires a lot more work for acceptance. In its current form, I recommend rejection but encourage the authors to submit an improved version to a future conference.

**Justification For Why Not Lower Score:**

N/A

---

### Decision · Program_Chairs · 2024-01-16

Reject